# Prospective Analysis of the Effectiveness of Targeted Botulinum Toxin Type A Injection Using an Ultrasound-Guided Single-Point Injection Technique for Lower Face Contouring

**DOI:** 10.3390/jcm13175337

**Published:** 2024-09-09

**Authors:** Hyun-Jung Ryoo, Ho Kwon, Jae-Seon Choi, Bo-Seong Sohn, Ja-Young Yoo, Hyung-Sup Shim

**Affiliations:** 1Department of Plastic and Reconstructive Surgery, St. Vincent’s Hospital, College of Medicine, The Catholic University of Korea, Seoul 16247, Republic of Korea; hjryoo92@gmail.com (H.-J.R.); 10cjs13@naver.com (J.-S.C.); 2Department of Plastic and Reconstructive Surgery, Uijeongbu St. Mary’s Hospital, College of Medicine, The Catholic University of Korea, Seoul 03312, Republic of Korea; kwonho@catholic.ac.kr; 3Santaclaus Aesthetic Clinic, Seoul 06120, Republic of Korea; apoptosis_sohn@hanmail.net; 4Ruby Clinic, SeoCho-Gu, Seoul 06267, Republic of Korea; wkdud4916@naver.com

**Keywords:** botulinum toxin, lower face contouring, masseter muscle, masseter hypertrophy

## Abstract

**Background**: Botulinum toxin type A (BoNT-A) injection is widely used for masseter hypertrophy. Traditional BoNT-A injection methods often incorporate landmark-guided blind injections, which approximate the shape of the masseter muscle inject across various points. Conversely, ultrasound (US)-guided injection techniques offer real-time visualization and dynamic monitoring, enhancing accuracy. **Patients and Methods**: 50 patients who underwent BoNT-A injections were included in this trial. One on the face side received a landmark-guided injection, and the other side was treated with a US-guided injection. Initial and post-procedure measurements of muscle thickness at the upper, middle, and lower regions were collected using ultrasound. **Results**: Both methods led to a significant reduction in muscle thickness one month after injection. In the upper area, the absolute difference in muscle thickness between the two methods was observed as a mean ± standard deviation (SD) value of 0.37 ± 0.0314 (*p* < 0.0001), indicating a superior effect with US-guided injection. Similarly, in the middle area, the mean ± SD difference was 0.41 ± 0.0608 (*p* < 0.0001) and in the lower area, the mean ± SD difference was 0.24 ± 0.0134 (*p* = 0.0004). **Conclusions**: This study demonstrated that the US-guided single-point injection technique is a more effective and accurate method for BoNT-A injection compared to the conventional method.

## 1. Introduction

The masseter muscle is one of the muscles for mastication. It originates from the zygomatic arch and inserts at the mandibular angle and ramus. Sometimes the masseter muscle becomes larger and looks aesthetically bad. Botulinum toxin (BoNT-A) injection of the masseter muscle is a widely used non-surgical option for lower face contouring. Particularly among East Asians with a squared jawline due to masseter hypertrophy, the cosmetic use of botulinum toxin (BoNT-A) injections continues to gain popularity for aesthetic enhancement [1]. In recent years, active research has led to a greater understanding of the mechanisms of botulinum toxin, along with established research outcomes on its long-term safety and efficacy. Consequently, its utility now extends beyond cosmetic purposes to steadily increasing applications for therapeutic purposes, such as migraine headaches, bruxism, temporomandibular joint disorders (TMDs), and dynamic rhytides [2,3].

The injection technique of BoNT-A for lower face contouring has been introduced through various methods in several articles. The commonly used injection method typically involves drawing reference lines on the lower face according to method specifications, identifying the anterior and posterior borders of the masseter muscle, and then administering injections at multiple points [4,5,6,7]. According to a recent systematic review article on the efficacy of BoNT-A injection techniques, the method recommended by the authors involves identifying the anterior and posterior borders of the masseter muscle, followed by injecting three points starting from the thickest part of the muscle. This technique is widely adopted and used [8]. On the other hand, except for cases of mild masseter hypertrophy ≤ 10 mm, the single-point injection method is not preferred due to its lower effectiveness in reducing muscle thickness and its potential for causing uneven muscle atrophy [9].

The previously mentioned BoNT-A injection methods can be described as landmark-guided blind injection techniques that indirectly predict the approximate shape of the masseter muscle, enabling consistent injections across multiple muscle points. Yet, these methods lack real-time monitoring of the actual position and depth of injection within the muscle during BoNT-A administration.

The ultrasound (US)-guided injection technique is gaining significant attention across various medical fields as it provides a real-time and dynamic display of injections. This method enhances the accuracy of medical injection techniques, ensuring stability and elevating efficacy. Furthermore, recent large-scale randomized controlled trials or meta-analyses have demonstrated that US-guided injections, compared to landmark-guided injections, exhibit improved accuracy, efficacy, and cost-effectiveness in musculoskeletal and joint structures as well as soft tissues [10]. In particular, US-guided steroid injections for carpal tunnel syndrome [11,12], US-guided interventions for joints such as the shoulder, elbow, and knee [13,14,15], myofascial trigger point blocks, and ganglion nerve blocks are actively being used [16,17].

However, there is a lack of research confirming injection dynamics using US-guided techniques for BoNT-A injections specifically aimed at facial contouring. Therefore, this study aimed to provide evidence regarding the efficacy and safety of BoNT-A injections for lower face contouring in patients with masseter hypertrophy. We sought to compare a US-guided single-point technique with a landmark-guided three-point blind technique, presenting a comparison of masseter volume reduction to establish evidence regarding efficacy and safety.

## 2. Patients and Methods

### 2.1. Patients

A prospective study involving the masseters of 50 individuals (27 men and 23 women, with a mean age of 38.7 years) was conducted between February 2020 and June 2024. Prior to participating, all patients provided written consent. The study received approval from the Institutional Review Board of St. Vincent’s Hospital (VC23RASI0088) and adhered to the principles outlined in the 1975 Declaration of Helsinki (revised in 2008). Eligible patients for enrollment were those with a complaint of bulky masseters and clinical hypertrophy diagnosed by a single plastic surgeon. Exclusion criteria included pregnancy, a history of drug allergy or any other serious medical illness, a history of prior BoNT-A injections in the masseter muscles, and any TMJ abnormalities confirmed by a dentist.

The pre-injection evaluation involved a morphologic assessment of each patient’s masseter hypertrophy status during resting and biting states through clinical examination using inspection and palpation. Furthermore, objective imaging evaluation was performed using clinical photos of the patient’s face in the anteroposterior view during resting and biting states, obtained from a distance of 1 m from the face, prior to the procedure.

The pre-procedural initial thickness (mm) values of the upper, middle, and lower parts of the masseter muscle were measured in real time using the Affiniti 50 (Philips Ultrasound Inc., Bothell, WA, USA) two-dimensional B-mode, providing a cross-sectional view. Additionally, the patient was instructed to bite down, and the thickest portion of the masseter muscle was identified on US during both contraction and relaxation states. Gel was applied to facilitate skin contact by the US probe during both pre-injection and post-injection examinations. However, during the injection procedure, an antiseptic liquid containing isopropyl alcohol and chlorhexidine gluconate was used for skin-to-probe contact.

Nabota^®^ (botulinum toxin type A; Daewoong Pharmaceutical, Seoul, Republic of Korea) was supplied in 50 IU of freeze-dried powder and reconstituted with 1 mL of saline solution. A 30-gauge needle and 1-cc syringe were used for BoNT-A injection. Left or right masseter muscles from each patient were randomly assigned for injection.

The follow-up was conducted one month after the injection. Clinical photographs were obtained in the same setting as before the procedure, and, using the same mode on US, the thicknesses of the upper, middle, and lower parts of the masseter muscle were assessed. Thickness-reduction percentages were measured for each location (upper, middle, and lower) relative to the pre-procedural thickness. Additionally, the occurrence of complications, such as hematoma or paradoxical masseter bulging, was also assessed.

### 2.2. Injection Method

#### 2.2.1. Landmark-Guided, Three-Point Injection Technique

Patients were instructed to clench their teeth, and the masseter muscles were palpated to identify both their anterior and posterior borders. Injections were administered at least 1 cm from both the anterior line and the tragus–mouth corner line (Figure 1) [8]. The initial injection was targeted at the thickest region of the masseter, with the two subsequent injections placed approximately 1–1.5 cm away, one each above and below the reference injection site. This approach ensured an even distribution of Nabota in the upper, middle, and lower regions of the masseter. In areas where the masseter’s thickest part measured >10 mm, Botox, prepared at a concentration of 50 IU/mL, was administered in a total volume of 1 cc, divided equally among the three injection points. This resulted in approximately 16 units being injected per point. In cases where the thickness was ≤10 mm, 0.5 cc of Botox at a concentration of 50 IU/mL was distributed among the three points, resulting in a total of 25 units administered, with approximately 8.33 units injected per point.

#### 2.2.2. US-Guided, Single-Point Injection Technique

The patient, who had previously received Botox injections in one masseter muscle using the landmark-guided, three-point injection technique, underwent injections on the opposite side of the face using the US-guided, single-point injection approach. Following the previously described method for masseter muscle assessment, the thickest region was identified using US B-mode, and its thickness (mm) was measured (Figure 2). Subsequently, with the US probe fixed in place, the injection needle was carefully guided into the muscle, ensuring proper and effective administration of the injection (Figure 3). Once it was confirmed that the injection was appropriately delivered at the thickest part of the muscle, the needle angle was adjusted to approximately 30 degrees in a thread-like manner to ensure even distribution of the drug throughout the remaining muscle parts (Figure 4). The needle was then withdrawn. The injection dosage remained consistent with the side that underwent landmark-guided, three-point injection.

### 2.3. Statistical Analysis

The data were analyzed using the paired t test to assess and compare the masseter thickness measured before and after injection by US. Statistical analysis was conducted using SPSS software (version 23.0 for Windows; IBM Corporation, Armonk, IL, USA). *p* < 0.05 was considered statistically significant.

## 3. Results

When comparing the landmark-guided injection group and the US-guided injection group, the pre-procedural upper-area masseter muscle thickness values were 9.92 mm and 9.93 mm, respectively, while those post-procedure were 8.99 mm and 8.63 mm. In the middle portion of the masseter, the preprocedural thickness for the landmark-guided injection group was 10.22 mm, while that in the US-guided injection group was 10.83 mm. Post procedure, these measurements were 8.72 mm and 8.92 mm, respectively. Finally, in the lower area, the landmark-guided injection group showed a pre-procedural thickness of 8.16 mm, while the US-guided injection group showed that of 8.43 mm. Post-procedure, these measurements were 7.49 mm and 7.52 mm, respectively (Table 1). After the procedure, no unusual complications were observed in either group.

Graphical comparison revealed a greater reduction in masseter muscle thickness in the US-guided injection group at all three regions compared to the landmark-guided injection group (Figure 5). When comparing the absolute difference in muscle thickness between two groups, the mean ± standard deviation (SD) difference in the upper area showed 0.37 ± 0.0314, the 95% confidence interval was 0.2893 to 0.4478, and the *p*-value was lower than 0.001. The mean ± SD in the middle area was 0.41 ± 0.0608 (*p*-value < 0.001), and the 95% confidence interval was 0.3472 to 0.4475. And in the lower area, the 95% confidence interval was (0.142, 1.258) with a mean ± SD difference of 0.24 ± 0.0134, and the *p*-value was 0.0004 (Table 2).

## 4. Discussion

Though the existing technique for lower face contouring using BoNT-A injection is inconsistent across studies, all are landmark-guided injection techniques. Among the prominent injection techniques, one method involves drawing a reference line connecting the tragus of the ear to the corner of the mouth, verifying the masseter muscle, and then injecting BoNT-A at four points along this line—two points above and two points below the reference line, spaced 1 cm apart [4]. In another study, a safety zone was defined by the boundaries of the earlobe–mouth corner line, the lower border of the mandible, and the anterior and posterior borders of the masseter. Within this safety zone, a three-point injection technique was introduced [6]. Kim et al. introduced a technique involving 3–4 injection points positioned 1.5 cm above the mandible angle [5]. Another study outlines a method of six evenly dosed injection points administered along the zygomatic arch to the region of the mandible angle while verifying the anterior and posterior margins of the masseter muscle [7].

Among several recently published systematic review articles on the efficacy of various BoNT-A injection techniques, Cheng et al. prefer and recommend a technique where the patient is instructed to bite down, enabling identification of the anterior and posterior borders of the masseter muscles. In their report, they introduce a technique involving three-point injections, starting from the thickest part of the muscle [8]. In the present study, we performed landmark-guided three-point injections in one masseter muscle of each patient, while the masseter on the opposite side of the face was treated with a US-guided single-point injection in a thread-like manner. Then, 1 month later, we assessed the reduction in masseter volume. Comparing the two sides in the same patient, the masseter volume significantly decreased post-treatment on the side where US-guided single-point injection was performed, except in the middle region, as shown in Table 2. In the case of landmark-guided injection, prior to the injection, the patient was instructed to bite down to stimulate the most prominent area of the masseter muscle, and the protruding point was used for assessment. Conversely, with US-guided injection, the thickness of the masseter muscle during injection was measured using the US thickness scale; this allows more precise identification of the thickest portion of the muscle, and the needle can be inserted directly into that location, enhancing injection accuracy.

In this study, regardless of the initial muscle thickness, the single-point injection approach led to a relatively greater masseter reduction effect compared to the three-point injection approach. Regardless of the specific region of the masseter, as evident from the graph, the slope of muscle thickness reduction suggests that the US-guided technique was more effective. According to a previous study, the lower one-third of the masseter muscle, often referred to as the “lower belly”, is the thickest part [18,19]. Although the thickest part of the masseter can vary among individuals, this lower portion tends to be relatively thicker due to overlapping of the deep inferior tendon (DIT) and the superficial belly. In this study, the reduction in thickness in the middle part showed a noticeable difference in the reduction slope.

A previous study outlined a botulinum toxin injection protocol based on masseter hypertrophy classification and recommends the single-point injection technique exclusively for individuals with mild (muscle thickness ≤ 10 mm), bulging type I (minimal bulging), or type II (mono bulging) hypertrophy. The use of single-point injection is discouraged in all other groups [9]. This recommendation stems from the potential risk of unintended adverse effects, such as non-uniform masseter atrophy or unnatural-appearing bulging. The literature supports multisite injections as a preferable alternative to address these concerns and to enhance the overall safety and efficacy of the procedure [8,18,19]. However, the method proposed in the present study uses US to monitor injection dynamics in real time, ensuring even distribution of the drug from the thickest part of the masseter to the rest of the muscle during the injection process. Consequently, uniform reductions in the upper, middle, and lower portions of the masseter were observed. Furthermore, compared to the three-point injection approach, the single-point method has the advantage of decreasing pain associated with needle puncture.

A notable advantage of US is its real-time visualization of structures such as joints, tendons, and nerves. Simultaneously, it is non-invasive and relatively cost effective, allowing wide use in outpatient settings. These advantages of US indicate it as a suitable imaging modality for musculoskeletal disorders [20,21]. Beyond its simple diagnostic purposes, when used as a guiding tool for musculoskeletal interventions, such as injection, aspiration, and nerve block, US enables real-time monitoring of injection dynamics. This capability allows operators to avoid critical adjacent structures and to propose methods to enhance the accuracy of targeted injections. As a result, the medical utility of US is increasing [22]. In particular, according to a meta-analysis of randomized controlled trials comparing landmark-guided and US-guided techniques for local corticosteroid injection in carpal tunnel syndrome, US-guided injections demonstrate superior outcomes in terms of symptom severity improvement [12]. Based on such evidence, the introduction of US in outpatient-focused local cosmetic clinics is valuable, as it allows assessment of muscle condition before procedures and visualization of injection dynamics during treatment and is useful in evaluating procedural responses during follow-up. Therefore, its versatility in these aspects is expected to be substantial.

The application of US in injection techniques for cosmetic purposes has been studied, particularly in the context of hyaluronic acid filler injections, for depth monitoring and prevention of vascular complications. According to a recently published study, US demonstrates aesthetic benefits as a tool enabling depth determination during filler injections, distribution of the pocket, and dynamic observation for long-term follow-up of filler changes over 180 days [23]. According to another study, for correcting the nasolabial fold using hyaluronic acid filler, a safe injection method involves identification of the facial artery using Doppler US. Notably, this approach involves the use of sonography to reduce vascular complications during injection [24]. In the field of cosmetic plastic surgery, BoNT-A injection is a widely used and valuable non-surgical technique for facial and neck contouring as well as wrinkle improvement across the entire face and neck.

While this study introduces a lower face contouring method focused on the masseter muscle, the incorporation of US, which allows direct monitoring and injection of various target muscles, is anticipated to lead to widespread market expansion in the field of non-surgical cosmetic interventions. Such use is supported by the proven high accuracy, stability, and cost-effectiveness of US in the present study.

A limitation of this study is the relatively small sample size. Future research should aim to include a larger number of subjects to achieve statistically significant results. Additionally, there is a need for follow-up studies to explore more efficient and effective methods, such as adjusting injection volumes based on masseter muscle thickness.

## 5. Conclusions

This study is a prospective comparison of US-guided single-point and landmark-guided three-point injections for BoNT-A injection in lower face contouring among 10 patients with masseter hypertrophy. The findings confirmed a greater reduction in masseter volume with the US-guided single-point technique compared to the use of landmark-guided three-point injections. Through this study, the US-guided single-point injection technique has been validated as an effective and safe method. It is anticipated that use of this technique in various target muscles will provide medical evidence for the expansion of both medical and cosmetic applications.

## Figures and Tables

**Figure 1 jcm-13-05337-f001:**
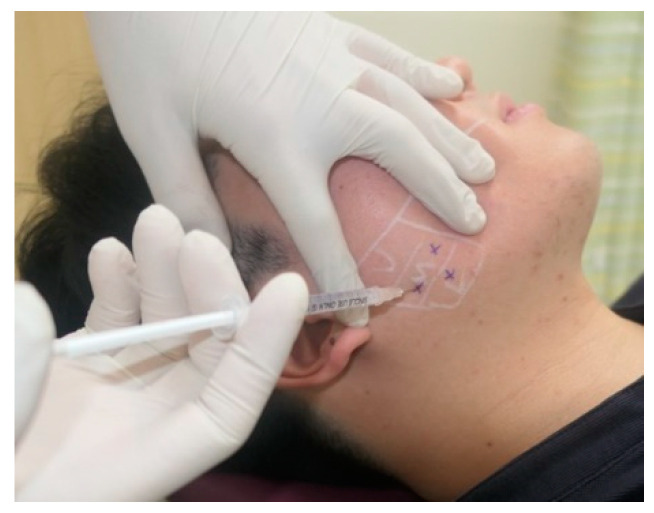
A landmark-guided BoNT-A injection procedure. The borders of the masseter muscle were identified by palpation, and BoNT-A first was injected at the thickest region, followed by two subsequent injections about 1–1.5 cm away.

**Figure 2 jcm-13-05337-f002:**
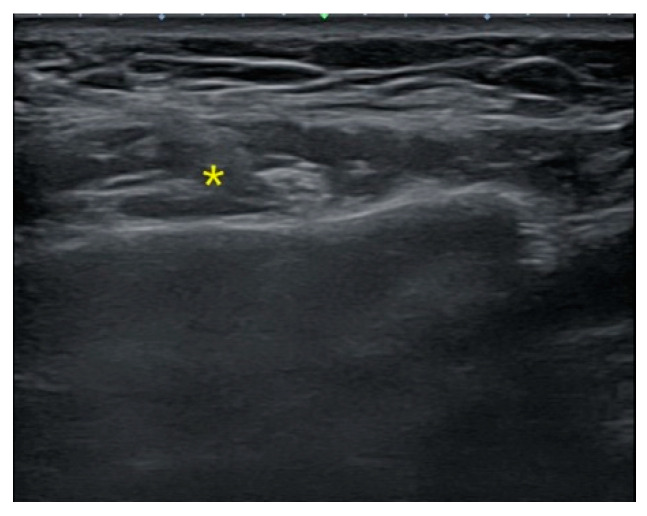
Ultrasound image before BoNT-A injection. Prior to the procedure, the thickest location of the masseter muscle was pre-identified using ultrasound. (* represents the thickest part of the masseter muscle).

**Figure 3 jcm-13-05337-f003:**
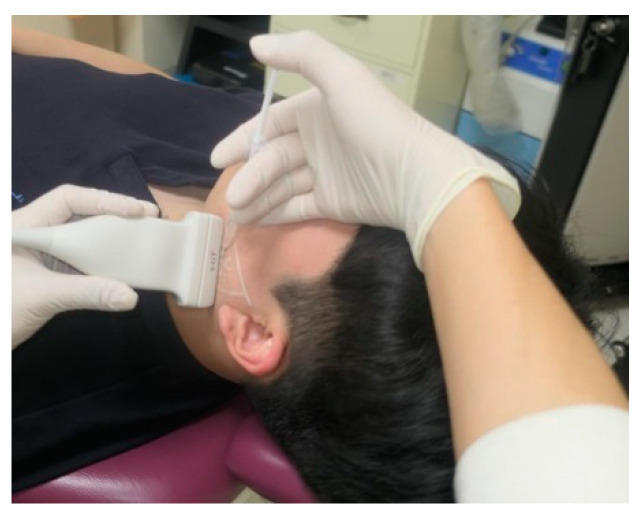
Ultrasound (US)-guided BoNT-A injection. With the US system set to B-mode, the thickest region was identified, and injections were administered in that specific area. Real-time monitoring using US confirmed the accurate delivery of injections.

**Figure 4 jcm-13-05337-f004:**
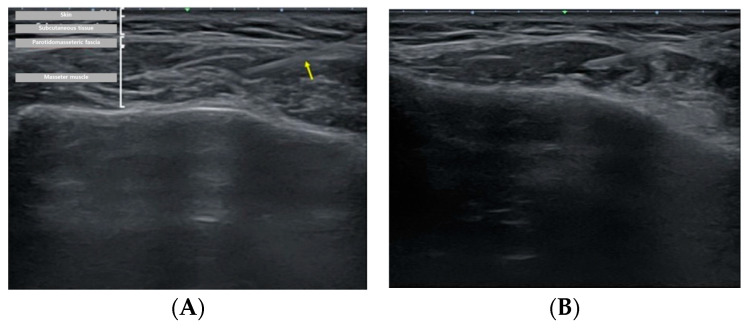
Ultrasound image showing injection dynamics after BoNT-A injection. (**A**) The yellow arrow notes the BoNT-A injection needle. Puncture was performed using a 1-cc syringe, with injection into the thickest region of the masseter muscle. (**B**) Immediate post-injection view.

**Figure 5 jcm-13-05337-f005:**
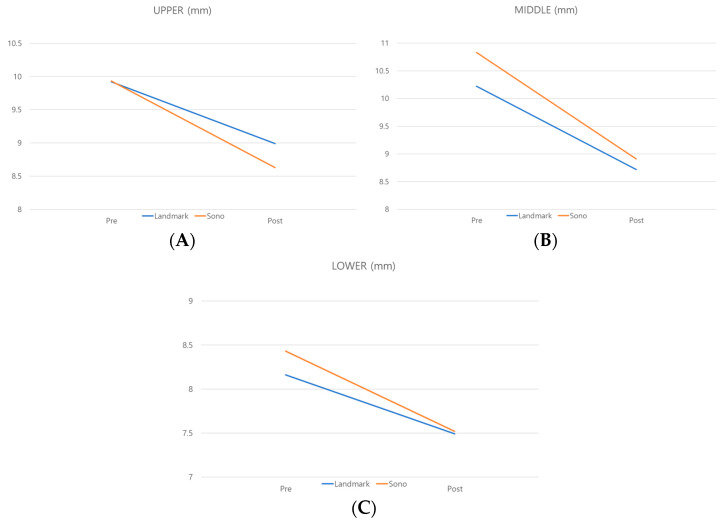
Changes in masseter muscle thickness before and after Botox injections. (**A**) In the upper area, masseter muscle thickness in the landmark-guided group measured 9.92 mm and 8.99 mm before and after the procedure, respectively, while that in the ultrasound-guided group measured 9.93 mm and 8.63 mm. (**B**) In the middle area, masseter muscle thickness values in the landmark-guided group were 10.22 mm and 8.72 mm before and after the procedure, respectively, while corresponding values in the ultrasound-guided group were 10.83 mm and 8.92 mm. (**C**) In the lower area, the masseter muscle measured 8.16 mm and 7.49 mm before and after the procedure in the landmark-guided group and 8.43 mm and 7.52 mm in the ultrasound-guided group, respectively.

**Table 1 jcm-13-05337-t001:** Masseter muscle thickness (mm).

	Landmark Guided		Ultrasound Guided	
	Pre	Post	Pre	Post
Upper	9.92	8.99	9.93	8.63
Middle	10.22	8.72	10.83	8.92
Lower	8.16	7.49	8.43	7.52

**Table 2 jcm-13-05337-t002:** Masseter muscle thickness (mm) difference 1 month after injection (U–L).

Site	95% Confidence Interval	Mean ± SD Difference	*p* Value
Upper	(0.2893, 0.4478)	0.37 ± 0.0314	<0.001
Middle	(0.3472, 0.4475)	0.41 ± 0.0608	<0.001
Lower	(0.1404, 0.3596)	0.24 ± 0.0134	0.0004

Abbreviations: L, landmark-guided three-point injection; U, US-guided single-point injection.

## Data Availability

The data underlying this article will be shared on reasonable request to the corresponding author.

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
