# Peer review of "Prospective Analysis of the Effectiveness of Targeted Botulinum Toxin Type A Injection Using an Ultrasound-Guided Single-Point Injection Technique for Lower Face Contouring"

_jcm, 2024, doi:10.3390/jcm13175337_

Round 1

Reviewer 1 Report

Comments and Suggestions for Authors

I have the following comments on the work:

During the initial assessment of the paper according to the guidelines of the JCM journal, numerous editorial oversights in the preparation of the work are immediately noticeable. I'll start with the most serious ones:

• The paper features a different order of authors in the MDPI system: 'Hyung-Sup Shim *, Hyun-Jung Ryoo, Ho Kwon, Jae-Seon Choi, Bo-Seong Sohn, Ja-Young Yoo' compared to the paper. This needs to be explained. It is a serious error.

Other editorial errors include:

• Above the title, there should be information about the type of article.

• Beside the names of the authors, please remove the academic titles – this journal does not use academic degrees.

• Line (L) – 12: of course, authors can indicate the same contribution. However, in the journal, this is marked with a different symbol. Quoting the JCM guidelines: 'Equal Contributions: authors who have contributed equally should be marked with a superscript symbol (†). The symbol must be included below the affiliations, and the following statement added: “These authors contributed equally to this work”.'

• Abstract: it exceeds the word count for original papers, which is 200.

• L58 – ‘(2, 3)’ – the citation format is square brackets at the end of the sentence before the period.

• At the end of the paper in the statements section, among other things, the Author Contributions and Funding sections are missing.

• The bibliography does not meet the journal's standards. The year should be bolded. Additionally, in works no. 20 and 24, full biographical information is missing.

• At the end of the paper, lines 365-367 are cut off.

Substantive errors and other questions:

• L50 – 'the masseter muscle' – please provide an introduction to the masseter muscle. Indicate that it is one of the chewing muscles – its location and function. Remember that specialists from various fields will read this work, so it is good to describe even such basic anatomical information.

• L57-58 – 'temporomandibular joint (TMJ) disorders' – this is not the correct abbreviation; it should be 'TMDs' after the word 'disorders'.

• L92 – what hypothesis did the authors propose?

• '10 individuals (5 men and 5 women)' – this is quite a small number, was the sample size calculated? Add information.

• Why was there no control group, e.g., subjected only to injections?

• L131-143 – is this the authors' methodology or did they base it on other works?

• 2.3 Statistical analysis – why was the T-test used on such a small group? Was a distribution analysis conducted at all?

• Additionally, effect size and confidence intervals should be added to the results. The p-value alone provides too little information.

I'm very sorry, but due to the small group, lack of control, and questionable statistics, I am currently stepping back from evaluating the work.

Author Response

During the initial assessment of the paper according to the guidelines of the JCM journal, numerous editorial oversights in the preparation of the work are immediately noticeable. I'll start with the most serious ones:

  • The paper features a different order of authors in the MDPI system: 'Hyung-Sup Shim *, Hyun-Jung Ryoo, Ho Kwon, Jae-Seon Choi, Bo-Seong Sohn, Ja-Young Yoo' compared to the paper. This needs to be explained. It is a serious error.
  1. A) The discrepancy in the order of authors between the paper and the MDPI system appears to be due to the configuration related to the corresponding author. In the paper, the correct order of authors has been established, placing the corresponding author at the end. Conversely, the MDPI system seems to be configured to position the corresponding author at the beginning.

Other editorial errors include:

  • Above the title, there should be information about the type of article.

An article was added above the title.

  • Beside the names of the authors, please remove the academic titles – this journal does not use academic degrees.

All academic titles have been removed.

  • Line (L) – 12: of course, authors can indicate the same contribution. However, in the journal, this is marked with a different symbol. Quoting the JCM guidelines: 'Equal Contributions: authors who have contributed equally should be marked with a superscript symbol (†). The symbol must be included below the affiliations, and the following statement added: “These authors contributed equally to this work”.'

The modifications have been made to comply with the requirements of your journal.

  • Abstract: it exceeds the word count for original papers, which is 200.

The abstract has been adjusted to 200 words.

  • L58 – ‘(2, 3)’ – the citation format is square brackets at the end of the sentence before the period.

The citations have been revised to conform to the formatting guidelines of your journal.

  • At the end of the paper in the statements section, among other things, the Author Contributions and Funding sections are missing.
    I have added the relevant information.
  • The bibliography does not meet the journal's standards. The year should be bolded. Additionally, in works no. 20 and 24, full biographical information is missing.

I have revised the references according to the JCM format. As for entries 20 and 24, all required details seem to be included. Could you please specify which information is missing?

  • At the end of the paper, lines 365-367 are cut off.

I have made the necessary corrections accordingly.

Substantive errors and other questions:

  • L50 – 'the masseter muscle' – please provide an introduction to the masseter muscle. Indicate that it is one of the chewing muscles – its location and function. Remember that specialists from various fields will read this work, so it is good to describe even such basic anatomical information.

An introduction to the masseter muscle has been added at the beginning of the article.

  • L57-58 – 'temporomandibular joint (TMJ) disorders' – this is not the correct abbreviation; it should be 'TMDs' after the word 'disorders'.

The revisions have been made in accordance with your requirements.

 L92 – what hypothesis did the authors propose?

We anticipated that a single injection using ultrasound (US) guidance, rather than multiple injections based on traditional landmarks, would achieve similar or even superior outcomes compared to the conventional method.

  • '10 individuals (5 men and 5 women)' – this is quite a small number, was the sample size calculated? Add information.

of the masseter muscle, we obtained 20 experimental results. A statistician advised us that this number is sufficient for statistical analysis. However, we acknowledge the concern that the absolute sample size may be limited and agree to gather more data in future studies.

  • Why was there no control group, e.g., subjected only to injections?

All participants underwent both landmark-based injection and ultrasound (US)-guided injection. This approach was intended to minimize bias and evaluate the effectiveness by accounting for the individual variations in the masseter muscle characteristics.

  • L131-143 – is this the authors' methodology or did they base it on other works?

In reviewing "Botulinum Toxin Type A for the Treatment of Masseter Muscle Prominence in Asian Populations" by Wan Wu, as well as other related studies and clinical applications, it is evident that the landmark-based injection method is a standard approach for treating typical cases of masseter hypertrophy. Although individual variations may occur in the details, the commonly used method is presented in section 2.2.1.

  • 2.3 Statistical analysis – why was the T-test used on such a small group? Was a distribution analysis conducted at all?

Before performing statistical calculations, we consulted with an expert who confirmed that statistical analysis for this study could be conducted using a T-test. Although there are 10 patients, each side of the face was counted separately, resulting in 10 data points per group and a total of 20 data points.

  • Additionally, effect size and confidence intervals should be added to the results. The p-value alone provides too little information.

Following your advice, I have included the confidence interval.

Reviewer 2 Report

Comments and Suggestions for Authors

Comments

The reviewer thinks the study is interesting and useful for readers. There were no critical faults in the study. The reviewer thinks it is worth being published after some revisions.

Injection Method

2.2.1 Landmark-guided, three-point injection technique

Why was the Botox (Allergan plc, Dublin, Ireland) injected? Why was not Nabota (botulinum toxin type A; Daewoong Pharmaceutical, Seoul, Korea) described before the section?  Typo?

2.2.2 US-guided, single-point injection technique

“The injection dosage remained consistent with the side that underwent landmark-guided, three-point injection.”

Was the dosage 50IU or 16IU in >10 mm, 25IU or 8.33IU in <10 mm?

3. Results

Please described if there were any post procedural complications in both groups.

In case there were several complications, please compare them between both groups.

The reviewer recommends the authors to show some photographs of representative cases in order to attract readers, if the regulations of the journal allows the number of figures.

4. Discussion

Please note some limitations of the study at the end of discussion.

Author Response

Injection Method

2.2.1 Landmark-guided, three-point injection technique

Why was the Botox (Allergan plc, Dublin, Ireland) injected? Why was not Nabota (botulinum toxin type A; Daewoong Pharmaceutical, Seoul, Korea) described before the section?  Typo?

- We acknowledge that the error was correct and have amended it to "Nabota."

2.2.2 US-guided, single-point injection technique

“The injection dosage remained consistent with the side that underwent landmark-guided, three-point injection.”

Was the dosage 50IU or 16IU in >10 mm, 25IU or 8.33IU in <10 mm?

-Regardless of the injection method, if the masseter muscle thickness was 10 mm or greater, a total of 50 units were administered. For thicknesses less than 10 mm, 25 units were injected.

  1. Results

Please described if there were any post procedural complications in both groups.

In case there were several complications, please compare them between both groups.

No unusual complications were observed in the patients. This information has been added to the results section.

The reviewer recommends the authors to show some photographs of representative cases in order to attract readers, if the regulations of the journal allows the number of figures.

Changes in the masseter muscle thickness were recorded solely in quantitative terms, and clinical photographs were not included. In future studies, we will ensure to incorporate this aspect into the manuscript. Thank you.

  1. Discussion

Please note some limitations of the study at the end of discussion.

I have added the limitations of this study in the discussion section.

Round 2

Reviewer 1 Report

Comments and Suggestions for Authors

In response to the authors' questions:

“I have revised the references according to the JCM format. As for entries 20 and 24, all required details seem to be included. Could you please specify which information is missing?”

Indeed, I made an error in publication 20 and in publication 24 – the volume and pages.

“• At the end of the paper, lines 365-367 are cut off. I have made the necessary corrections accordingly.” – This comment has not been addressed.

In response to my main concerns:

“of the masseter muscle, we obtained 20 experimental results. A statistician advised us that this number is sufficient for statistical analysis. However, we acknowledge the concern that the absolute sample size may be limited and agree to gather more data in future studies.” – I do not accept the authors' response as they did not present the appropriate calculations.

“• 2.3 Statistical analysis – why was the T-test used on such a small group? Was a distribution analysis conducted at all? ‘Before performing statistical calculations, we consulted with an expert who confirmed that statistical analysis for this study could be conducted using a T-test. Although there are 10 patients, each side of the face was counted separately, resulting in 10 data points per group and a total of 20 data points.’” – I do not accept the response. With all due respect, the fact that someone said something cannot be considered substantive evidence. I have concerns about this point as a statistician, and my concerns have not been addressed.

“• Additionally, effect size and confidence intervals should be added to the results. The p-value alone provides too little information. Following your advice, I have included the confidence” – I do not see in the manuscript where this has been added.

With all due respect to the authors, I maintain my decision due to the lack of satisfactory responses to my concerns.

Best regards,

Author Response

I made an error in publication 20 and in publication 24 – the volume and pages.

reply: I have made the revisions according to your comments.

lines 365-367 are cut off

reply: While revising the manuscript, I thoroughly reviewed the entire document, and there were no cut-off sections.

I do not accept the authors' response as they did not present the appropriate calculations,  I do not accept the response. With all due respect, the fact that someone said something cannot be considered substantive evidence. I have concerns about this point as a statistician, and my concerns have not been addressed.

reply: In order to ensure statistical significance, we conducted an additional investigation with 40 more patients, bringing the total number of subjects to 50. For each group, we conducted the Shapiro-Wilk normality test and confirmed the p-value to validate the normal distribution, after which we performed a T-test. All results, including confidence intervals, have been presented.

Reviewer 2 Report

Comments and Suggestions for Authors

The reviewer is satisfied with the revision

Author Response

Thanks for your assignment.